# A Newly Established Murine Cell Line as a Model for Hepatocellular Cancer in Non-Alcoholic Steatohepatitis

**DOI:** 10.3390/ijms20225658

**Published:** 2019-11-12

**Authors:** Andreas Kroh, Jeanette Walter, Herdit Schüler, Jochen Nolting, Roman Eickhoff, Daniel Heise, Ulf Peter Neumann, Thorsten Cramer, Tom Florian Ulmer, Athanassios Fragoulis

**Affiliations:** 1Department of General, Visceral and Transplantation Surgery, Uniklinik RWTH Aachen, 52074 Aachen, Germany; jeanette.walter@rwth-aachen.de (J.W.); jnolting@ukaachen.de (J.N.); reickhoff@ukaachen.de (R.E.); dheise@ukaachen.de (D.H.); uneumann@ukaachen.de (U.P.N.); tcramer@ukaachen.de (T.C.); fulmer@ukaachen.de (T.F.U.); afragoulis@ukaachen.de (A.F.); 2Institute of Human Genetics, Uniklinik RWTH Aachen, 52074 Aachen, Germany; hschueler@ukaachen.de; 3Department of Surgery, Maastricht University Medical Center, 6200 MD Maastricht, The Netherlands; 4ESCAM—European Surgery Center Aachen Maastricht, 52074 Aachen, Germany; 5ESCAM—European Surgery Center Aachen Maastricht, 6200 MD Maastricht, The Netherlands; 6Department of Anatomy and Cell Biology, Uniklinik RWTH Aachen, 52074 Aachen, Germany

**Keywords:** non-alcoholic steatohepatitis, hepatocellular carcinoma, cell line, mTOR, Everolimus, KU-0063794

## Abstract

Non-alcoholic steatohepatitis (NASH) has become a major risk factor for hepatocellular cancer (HCC) due to the worldwide increasing prevalence of obesity. However, the pathophysiology of NASH and its progression to HCC is incompletely understood. Thus, the aim of this study was to generate a model specific NASH-derived HCC cell line. A murine NASH-HCC model was conducted and the obtained cancer cells (N-HCC25) were investigated towards chromosomal aberrations, the expression of cell type-specific markers, dependency on nutrients, and functional importance of mTOR. N-HCC25 exhibited several chromosomal aberrations as compared to healthy hepatocytes. Hepatocytic (HNF4), EMT (Twist, Snail), and cancer stem cell markers (CD44, EpCAM, CK19, Sox9) were simultaneously expressed in these cells. Proliferation highly depended on the supply of glucose and FBS, but not glutamine. Treatment with a second generation mTOR inhibitor (KU-0063794) resulted in a strong decrease of cell growth in a dose-dependent manner. In contrast, a first generation mTOR inhibitor (Everolimus) only slightly reduced cell proliferation. Cell cycle analyses revealed that the observed growth reduction was most likely due to G_1_/G_0_ cell cycle arrest. These results indicate that N-HCC25 is a highly proliferative HCC cell line from a NASH background, which might serve as a suitable in vitro model for future investigations of NASH-derived HCC.

## 1. Introduction

Hepatocellular carcinoma (HCC) is the third leading cause of cancer-related deaths worldwide [1]. Therapeutic options vary from surgical therapy and percutaneous ablation under curative intent to cytostatic therapy and transarterial chemoembolisation (TACE) under palliative intent [2,3,4]. The majority (70%) suffers from metastasis or de novo HCC, despite a five-year survival rate of 70% in patients with a curative treatment strategy [5,6,7,8]. The high recurrence rate of HCC is especially based on the survival of cancer stem cells (CSCs), which are a subpopulation of cells that are resistant to chemotherapy and radiation [9]. Therefore, CSCs are a current focus in HCC research. Non-alcoholic steatohepatitis (NASH) has become the most frequent cause of HCC [10] due to the rapid increase of obesity in the past decades and available therapies for the other relevant HCC risk factors hepatitis C and B (HCV and HBV) [11,12]. The development of both HBV/HCV- as well as NASH-derived HCC is not yet fully understood. Although they share some common hallmarks (inflammation, fibrosis, cirrhosis), the pathogenesis and outcome still differ in several aspects. By escaping the HLA-II-mediated immune response, HBV/HCV infects hepatocytes, whereby oxidative stress and the subsequent recruitment of inflammatory cells activate hepatic stellate cells (HSC) and collagen deposition. Besides, several HCV proteins are also known to induce fibrogenic and inflammatory processes of HSCs [13,14]. In viral hepatitis, initial fibrosis mostly develops within portal tracts, which leads to persistent portal inflammation, cirrhosis, and ultimately HCC. In contrast, dyslipidaemia, a commonly observed phenomenon in obese patients suffering from non-alcoholic fatty liver disease (NAFLD), leads to increased hepatic lipid deposition (steatosis) and more generalized liver inflammation. This, in turn, triggers a copious development of liver fibrosis, which causes NASH that can advance to HCC either via cirrhosis or without. [15,16,17,18]. However, NASH occurs not only in obese, but also in non-obese patients [15], which indicates a multifactorial genesis. Besides fibrosis, the typical hallmarks of NASH-derived HCC in patients are metabolic changes, such as weight gain, insulin resistance, diabetes mellitus, and triglyceridemia, as well as histopathological changes, such as hepatocyte ballooning, steatosis, lobular inflammation, and Mallory Denk bodies [19]. So far, the therapeutic options include lifestyle modification, treatment with vitamin E or pioglitazone, as well as bariatric surgery [20,21]. While the required weight loss of 10% for the reduction of inflammation and fibrosis is difficult to achieve, treatment with vitamin E and pioglitazone failed to reduce fibrosis and displays numerous long-term side effects. Currently, several clinical phase II/III trails are evaluating the benefit of e.g., antifibrotic, lipidaltering-, and anti-inflammatory targets [21].

Suitable in vitro models are needed in order to enable the investigation of NASH-derived HCCs. In this study, we provide a highly proliferative hepatocellular cancer cell line from mice with NASH-derived HCC, named N-HCC25. As HCC is a highly heterogeneous cancer type [22], it was essential to characterize the newly isolated cell line regarding chromosomal aberrations, expression of cell type-specific markers, cell metabolism, and growth. The dependency of N-HCC25 on nutrients was explored, as the metabolism of glucose, glutamine and fetal bovine serum (FBS) are currently discussed as targets for future HCC therapy [23,24,25,26]. As the mechanistic target of rapamycin (mTOR) is a major regulator of cell growth and metabolism, which is upregulated in 40–50% of HCCs [27,28,29], it is frequently addressed in HCC research [30]. While mTOR complex 1 (mTORC1) responds to nutrients, growth factors, stress, energy status, and amino acids, mTOR complex 2 (mTORC2) only responds to growth factors and is insensitive to nutrients. mTORC1 promotes cell growth, cell cycle progression, metabolism, protein biosynthesis, lipogenesis and negatively regulates autophagy [31]. Besides metabolism and cell growth, mTORC2 is mainly involved in cell survival [32,33] and cytoskeletal organization [34,35]. First generation mTOR inhibitors, also known as rapalogs, suppress mTORC1; second generation inhibitors block either mTORC1/mTORC2 or mTORC1/PI3K. Despite a significant anti-proliferative effect in preclinical studies [36,37] and success in the therapy of other solid tumors [38], no survival benefit was shown for single agent first generation inhibitor treatment of HCC in clinical studies [39]. Currently, these rapalogs are being investigated in combination with other drugs in patients with advanced HCC [38]. In contrast, second generation inhibitors have shown a stronger cytotoxic effect than rapalogs in preclinical studies [40,41,42,43,44]. In this study, the first generation inhibitor Everolimus and the second-generation inhibitor KU-0063794 were used to investigate the effect of mTOR inhibition on NHCC25. While Everolimus currently serves as an additive in clinical studies for advanced HCC and as adjuvant therapy in HCC patients after liver transplantation or TACE to prevent HCC recurrence [38], the growth inhibiting effects of the highly specific mTORC1/mTORC2 inhibitor KU-0063794 [45] were shown for HepG2 both in vitro and in vivo [46]. As investigation of NASH-derived HCC is required for sufficient future HCC therapy, we provide a new isolated and characterized HCC cell line from a murine NASH-HCC model.

## 2. Results

### 2.1. Isolation of N-HCC25 Cell Line

A modified version of the mouse model originally described by Yoshimoto, Loo et al. [47] was used to investigate NASH-derived HCC on a cellular level (Figure 1a). The treated animals displayed characteristic pathological changes, as also seen in human NASH-HCC, such as significant weight gain, insulin resistance, and glucose intolerance (data not shown). NASH was further confirmed by histopathological evaluation with both NAS (NASH-HCC = 4 compared to 7,12-Dimethylbenz[a]anthracene (DMBA) control = 1) [48] and SAF (NASH-HCC = 6 S_2_A_2_F_2_ compared to DMBA control = 1 S_0_A_1_F_0_) [49] Score (as is used in the human situation). Representative images of H&E, PAS as well as Sirius Red staining are shown in the Appendix A. The occurring tumours were visible macroscopically and detectable via in vivo µCT (Figure 1b). After 34 weeks, the N-HCC25 cells were isolated from the liver. Figure 1c shows representative pictures of early and late passages of N-HCC25 cells.

### 2.2. N-HCC25 Cells Exhibit Chromosomal Aberrations Associated with HCC Pathology

The origin of N-HCC25 cells was studied via cytogenetic analyses comparing primary hepatocytes and N-HCC25 cells. Our results demonstrate that primary murine hepatocytes were all diploid (Appendix A), while NHCC25 were polyploid and they accumulated chromosomal instability to a considerable extent. 

The number of chromosomes decreased with the number of cell culture passages, whereby certain chromosomes or chromosomal segments remained relatively stable (e.g., MMU X D-F4, 1 A-F, 11 B-E2, 6, 8), while others seemed to be preferentially eliminated (e.g., MMU Y, 3, 4, 7, 10, 14, 18) (Table 1).

The degree of polyploidy of the cells could not be determined with certainty, but it was presumed that the majority of cells probably had a tetraploid state. In the early cell culture passage, the average number of chromosomes was 65 (based on 25 mitoses, range: 60–69 chromosomes) with penta- and tetrasomeric status (5- and 4-fold occurrence, respectively) of several chromosomes. In the later passage, the average number of chromosomes decreased to 56 (26 mitoses, range 50–59), whereby 12% of the cells revealed no (penta- or) tetrasomic status of any chromosome.

Besides the changes in copy number and an aneuploidy rate of 100% in the cells of early and late passage, recurrent rearrangements, which become stable during time (passages), were also observed (Figure 2a–d, Table 2).

An unbalanced translocation between chromosome 1 and 11 with breakage at band 1 F and band 11 B, respectively, was found. This aberration was finally detected in every metaphase, which resulted in copy number gain of the segment 11 B-E2.

By molecular cytogenetic (FISH) analyses, an unbalanced X-autosome translocation of one of the X chromosomes with involvement of the segment A2 to F3 of chromosome 15 was detected. A possible spreading-of-inactivation of the X chromosome might silence a portion of the adjacent autosomal segment of chromosome 15.

In the early passage, 73% of the cells had a Robertson fusion between one of the chromosomes 16 and 19 (Rb16.19), which formed a metacentric chromosome. The proportion of cells with Rb.16.19 rose to 86% in the later passage. Robertson fusions involving chromosome 19 are rarely observed. This underrepresentation of chromosome 19 is attributed to the particular gene endowment in the proximal region.

A deletion of chromosome 17 (Del (17)) was consistently detectable in the cells of the late passage 29 and it caused a loss of the segment E1-E5. 

FISH analyses uncovered the loss of the Y chromosome in 44% and 55% of the cells, respectively. However, a significant portion of interphase nuclei showed a marked amplification of the Y chromosome (Appendix A). 

A list of the genes that may be affected by the above-mentioned aberrations, including tumor-drivers/oncogenes (gain-of-function), tumour-suppressors (loss-of-function), as well as microRNAs (both), are supplied in the Appendix A.

### 2.3. N-HCC25 Cells Derive from Hepatocytes and Express Cancer-Related Markers

The origin and characteristics of N-HCC25 were further explored on the transcriptional level in absence/presence PCR experiments (Figure 3). Early and late passages were analyzed to include early changes in gene expression. While N-HCC25 cells did not show expression of Albumin as a basic marker for hepatocytes, the gene expression of HNF4 was still found in the early passages P5–10. Specific HCC tumor markers, such as CK19, Sox9, and EpCAM, were expressed in all analyzed passages. Moreover, P5–24 showed the expression of CD44, which is known as a common cancer stem cell marker. The epithelial mesenchymal transition markers Twist and Snail were also expressed in all of the observed passages of N-HCC25. 

### 2.4. Growth of N-HCC25 Cells Mostly Depends on Sufficient Levels of Glucose and FBS

In tumor cells, the availability of basic nutrients plays an important role for metabolism and proliferation. In starvation experiments of N-HCC25, no statistically significant effects were found after an incubation time of 24 h with reduced glutamine, FBS, or glucose levels (Figure 4a–c). In contrast, a significant decrease in cell count was observed in cells that were cultured with 0.5 g/L and 0 g/L glucose for 48 h (Figure 4c). Equally, the cell count was significantly reduced in cells that were cultured in FBS- or glutamine-free culture medium for 48h (Figure 4a,b). The effect of starvation was further investigated by xCELLigence real-time cell analyzer (RTCA) that enables the longitudinal measurement of cell density in real-time. During cell adherence, no differences in proliferation occurred between the experimental groups (Figure 4d,e, phase I; Figure 4f, t_1_). Full medium control (FM) and cells that were cultured in reduced glutamine initially showed rapid proliferation until reaching a plateau 48 h after stimulation (Figure 4d, phase II), as indicated by high slopes and cell index (CI) (Figure 4e,f, phase II or t_2_ resp., 1st and 2nd column). In comparison, the treatment of N-HCC25 cells with reduced FBS and glucose resulted in a lower proliferation index (Figure 4d–f, phase II or t_2_ resp., 3rd and 4th column). While treatment with reduced FBS led to a maximum cell density of 2.7 ± 0.3 CI 24 h after stimulation, which lasted until the end of the experiment, (Figure 4d, phase II–III; 4f, t_2_-t_3,_ 3^rd^ column), cells that were treated with reduced glucose initially showed rapid growth followed by a massive decrease in cell density from 36 until 84 h after stimulation. In phase III, all of the experimental groups showed little cell proliferation (Figure 4e, phase III) and cell indexes of t_3_ basically corresponded to the values shown in t_2_ (Figure 4f).

### 2.5. N-HCC25 Cells Exhibit mTOR Activity, which Can Be Blocked by Specific Inhibitors

The influence of first generation (Everolimus) and second generation (KU-0063794) mTOR inhibitors on the mTOR pathway in N-HCC25 cells was explored, as mTOR is a major regulator of cell growth and a target for pharmacological treatment of HCC in clinical studies. As shown in Figure 5, all experimental groups expressed mTOR. Its autophosphorylation side Ser2481, which is mainly found in the mTOR complex 2, was more phosphorylated in controls and after 6 h of incubation with Everolimus, but less after 6 h of treatment with KU-0063794 and 24 h with both inhibitors. The phosphorylation of mTOR at Ser2448 by AKT serves as a marker for mTORC1 activity. While mTOR was more phosphorylated at Ser2448 in FM and DMSO control, phosphorylation was clearly reduced after treatment with the inhibitors. The rapamycin-insensitive companion of mTOR, named Rictor, was weakly expressed in all of the experimental groups. The expression of GßL (also known as LST8), which is comprised in both mTOR complexes, was shown in all experimental groups, but it was weakly expressed after treatment with Everolimus or KU-0063794 for 24 h. The ribosomal protein S6 and 4E-BP1 are both downstream targets of mTORC1, whose phosphorylation indicates mTORC1 activity. Phosphorylated forms are both only present in FM and DMSO control, but not in cells that were treated with mTOR inhibitors. 

### 2.6. mTOR Inhibition Leads to Decreased Proliferation of N-HCC25 Cells

Next, RTCA was used to compare the effect of Everolimus and KU-0063794 (1, 2.5, and 5 µM each) on cell proliferation in a longitudinal manner. During cell adherence, no differences in proliferation were found between the experimental groups (Figure 6a–f, phase I or t_1_ resp.). While FM and DMSO control initially showed a high increase of CI (Figure 6a,b, phase II, 1st and 2nd column), cells treated with different concentrations of Everolimus (Figure 6a,b, phase II, 3rd to 5th column) proliferated less than controls. The opposite effect was found in phase III, as indicated by an increased slope in cells that were incubated with the first generation mTOR inhibitor (Figure 6b, phase III, 3rd to 5th column). However, no significant differences were found between the CI values of the experimental groups at the timepoints t_2_ and t_3_ (Figure 6c). Cells that were treated with KU-0063794 also showed an initially slower growth as compared to FM and DMSO control (Figure 6d, phase II). Treatment with the second generation mTOR inhibitor reduced cell growth in a concentration-dependent manner (6d-f, phase II or t_2_ resp., 3^rd^ to 5^th^ column). In phase III, all of the groups that were treated with KU-0063794 exhibited a faster growth than FM and DMSO control, as indicated by a higher slope (Figure 6e, phase III). At both timepoints t_2_ and t_3_, increasing concentrations of the second generation mTOR inhibitor led to a significantly lower cell density as compared to FM control (Figure 6f). The strongest effect was found in treatment with 5 µM KU-0063794 (Figure 6f, phase III, 5^th^ column).

### 2.7. mTOR Pathway Inhibition Drives N-HCC25 Cells Into G_1_/G_0_ Cell Cycle Arrest

Cells were stained with anti-BrdU antibody, as well as PI, and analyzed by flow cytometry to analyze the influence of first and second generation mTOR inhibitors on cell cycle distribution. PI^low^BrdU^low^ cells were assigned to G_1_/G_0_ phase, PI^+^BrdU^high^ to S and PI^high^BrdU^low^ to G_2_/M respectively. When compared to FM and DMSO control, cells that were treated with Everolimus or KU-0063794 showed a significant increase of G_1_/G_0_ phase and a significant decrease in S phase with a higher magnitude for KU-0063794 (Figure 7a–c). While these effects were shown in a concentration dependent manner for KU-0063794, no statistically significant effects were found between the different concentrations of Everolimus. The number of cells in G_2_/M phase was not altered in any experimental group as compared to controls.

## 3. Discussion

NASH has become the major risk factor for HCC due to a worldwide increasing prevalence of obesity [12]. In the current study, a model specific NASH-HCC cell line was generated to investigate the biology of NASH-derived HCC [47]. This animal model resembled the human pathology in major metabolic (weight gain, insulin resistance, and glucose intolerance) and histopathological (steatosis, hepatocyte ballooning, inflammation, fibrosis) characteristics (Appendix A) [19]. The grade of NASH was determined by the widely used NAS and SAF Scores [48,49].

Cytogenetic analyses revealed numerical as well as structural aberrations in N-HCC25 cells that can be observed in HCC or cancer in general. Prominent chromosomal rearrangements were found besides the fundamental increase of ploidy grade, which is not necessarily unusual for hepatocytes, but might still be associated with gain-of-function. In this context, unbalanced translocation of chromosome 11, deletion of chromosome 17, robertsonian fusion of chromosome 16 and 19, as well as the occurrence of double-minutes (r/dmin/cen) was of special interest. In silico analyses of affected genomic content revealed several genes that might be interesting in the context of HCC and other cancer entities. The investigation of potential gain-of-function of the genes located on amplified genomic content (tetrasomy, unbalanced translocation of chromosome 11) as well as loss-of-function of genes that are affected by deletion (chromosome 17) or spreading-of-inactivation by the unbalanced X-autosome translocation would be interesting. However, this would be too extensive for this study and will be addressed in our upcoming project. In tumor cells, the analysis of gene expression allows for conclusions regarding their origin and prognosis. Here, the expression of hepatocytic nuclear factor 4 (HNF4) in early passages indicated that N-HCC25 cells were derived from hepatocytes [50]. HNF4 is considered as protective factor due to its inhibiting effect on fibrosis and tumorigenesis of HCC by inhibition of ß-catenin signalling [50,51]. External application of the transcription factor on HCC cell lines inhibited cell proliferation by differentiation of cancer cells to hepatocytes [9]. 

In the course of tumorigenesis, cancer cells undergo epithelial mesenchymal transition (EMT) [52], whereby epithelial markers are lost and mesenchymal features are gained [53]. The simultaneous expression of the EMT initiator CD44 [54] and the EMT markers Twist and Snail [55] indicate that EMT took place in N-HCC25. This process possibly resulted in the absence of the hepatocytic marker albumin and the loss of HNF4 in late passages.

In HCC, the loss of HNF4 [9,51,56] and expression of the cancer stem cell (CSC) markers CD44, CK19, Sox9, and EpCAM are considered to be indicators for poor prognosis [57,58,59,60]. In N-HCC25, these CSC markers were expressed in all passages. Previous studies characterized HCC with the expression of the cholangiocytic markers, such as CK19 and Sox9, as especially aggressive and, therefore, suggested defining a HCC with cholangiocellular differentiation [61,62]. EpCAM is considered to be a CSC marker, since its intracellular part EpICD mediates anchorage and growth factor independent cell proliferation, which is a typical trait in cancer [63]. It is crucial to assess the activity of the molecule rather than its presence, as the adhesion molecule is regularly expressed in the junctional zone of epithelial cells and upregulated in cancer cells. As only shown in the absence/presence of PCR, the activity of EpCAM in N-HCC25 was not assessed. In summary, the results of the gene expression analysis indicate that N-HCC25 is a highly aggressive hepatocellular cancer cell line. 

Recently, several findings have suggested that targeting tumor metabolism, which fundamentally differs from normal cells, might be a successful strategy for anticancer therapy [64]. For instance, cancer cells show little dependency on growth factors, but high dependency on energy supply [65,66]. In general, nutrient deficiency might cause G_1_/G_0_ arrest, if nutrients are required for multiplication, and might result in cell death, if nutrients are required for maintenance of G_1_/G_0_ phase [67]. In our study, the growth of N-HCC25 was significantly limited under treatment with halved amounts of FBS as compared to full medium, but did not lead to cell death. This indicates that N-HCC25 depended on FBS as a source for growth factors, despite the expression of EpCAM. This goes along with the finding of Zhou, Tang et al., who demonstrated that five human hepatoma cell lines showed restricted growth under FBS deprivation [68]. 

Tumor cells are also known for their excessive use of glucose via aerobic glycolysis (the so-called Warburg effect) [69]. Moreover, the expression of the glucose transporter GLUT1 increases during malignant transformation of hepatocytes and thus enables more energy supply and protein biosynthesis for mitosis [26]. In our study, the high dependency of N-HCC25 on glucose was confirmed. After an initial period of decelerated growth, glucose deprivation down to 50% resulted in cell death, as indicated by a sharp decrease of CI, although glutamine was still available as energy supply. Interestingly, a slight increase of CI was observed 84 h after glucose deprivation. As the impedance that is given in CI cannot be equated with cell count, the increase either might have been induced by changes in cell morphology or reinstating cell growth. The latter may be enabled by survival of tumour initiating cells (TICs), which express more GLUT1 and 3 and can therefore survive glucose deprivation [25]. Taken together, our findings support the idea that limiting glucose supply might be a promising approach in anticancer therapy.

Glutamine serves as energy supply, enables DNA synthesis, reduces reactive oxygen species (ROS), and supports redox homeostasis [70]. Therefore, the deprivation of glutamine might inhibit cell growth [71] or lead to cell death. In N-HCC25 cells, the deprivation of glutamine to 50% only slightly affected cell proliferation in RTCA. The small effect in RTCA might have occurred because the still available amount of glutamine was a sufficient supply since treatment with glutamine-free medium for 48 h led to a significant decrease in cell growth. As restriction of glucose supply led to cell death, while glutamine restriction only slightly affected cell growth, it can be hypothesized that sufficient glucose supply ensured N-HCC25 survival, despite glutamine restriction. Furthermore, glutamine deprivation promotes a more efficient metabolism of glucose by oxidative phosphorylation instead of anaerobic glycolysis or biosynthetic pathways in order to enable cell survival [23]. Beyond that, two out of six HCC subgroups show a mutation in the ß-catenin gene, leading to an overexpression of glutamine synthetase [22]. However, the mutation might either increase or attenuate glutamine dependency [24,72]. In HCV-HCC patients, glutamine inhibitors are applied to reduce fibrosis and inflammation, which both play an important role in pathophysiology of NASH [73]. Therefore, glutamine will remain an interesting target in NASH-HCC research.

As HCC is a heterogeneous tumor, various dysfunction of cellular processes and signalling pathways have been identified to be involved in the development of this cancer. These include alterations of telomerase reverse transcriptase (TERT), the Wnt pathway, JAK-STAT pathway, DNA repair mechanisms by TP53, chromatin remodelling by SWI/SNF, as well as the PI3K-AKT-mTOR pathway [74], all of which are investigated as possible targets for anticancer therapy. A clinically already addressed target is the mTOR pathway, which plays a major role in metabolism and cell growth. The upregulation of mTOR occurs in 40–50% of HCCs [27,28,29] and it is associated with low differentiation, poor prognosis, and early recurrence [29,75,76]. Despite preclinical success, so far limited clinical profit was achieved by the inhibition of mTOR. The first generation inhibitor Everolimus is known for its antineoplastic effects, which are mainly achieved due to the inhibition of proliferation and angiogenesis [77,78]. In several randomized controlled trials, survival benefits have been reported for solid cancers [79,80,81]. Regarding HCC, Everolimus is used in clinical studies for patients with advanced HCC, after liver transplantation and TACE [82,83,84,85]. Synergistic effects have been described for the application of Everolimus and calcineurin inhibitors to avoid HCC recurrence after liver transplantation [84,85] and for the combination of Everolimus with radiation [86]. However, low response rates have been reported in patients that were treated with single-agent Everolimus [79,80,81]. Additionally, preclinical studies did not show dose-dependent Everolimus’ antitumor effects [82,87,88]. Our findings support this observation, as the deceleration in cell growth that was observed in RTCA was not dose-dependent for cells that were treated with Everolimus.

Cell cycle progression is promoted by mTORC1 [31]. Correspondingly, a decrease in cell cycle progression was found in N-HCC25 cells that were treated with Everolimus or the second generation inhibitor KU-0063794. Interestingly, the inhibiting effect of KU-0063794 was dose-dependent and it exceeded the effect of Everolimus, which suggests that KU-0063794 has a stronger effect on mTORC1 than Everolimus. This goes along with the finding of Garcia-Martinez that KU-0063794 leads to a greater dephosphorylation of the mTORC1 downstream target 4EBP1 than rapamycin [45]. The increased effect on mTORC1 might be a reason for the greater effects of KU-0063794 when compared to Everolimus. This circumstance is supported by the fact that mTORC2, also inhibited by KU-0063794, is considered to be responsible for cell survival [31].

Recently, investigators found out that autophagy, which is normally inhibited by mTORC1 [31], is upregulated in cells that were treated with single-agent Everolimus or KU-0063794 and promotes cell survival rather than cell death [89,90]. This is supported by the finding that cytotoxity in KU-0063794 treated cells is enhanced when autophagy is simultaneously inhibited [46]. Interestingly, Lee et al. described a synergistic cytotoxic effect of Everolimus and/or KU-0063794 by the inhibition of autophagy via a decrease of the positive autophagy regulator SIRT [89]. Taken together, mTOR remains an important target for research in HCC and especially the knockout of both complexes will play a role in future research.

In conclusion, N-HCC25 cells show characteristics of a highly proliferative NASH-derived hepatocellular carcinoma cell of robust malignancy and might, therefore, serve as a suitable in vitro model for future investigations of NASH-derived HCC.

## 4. Materials and Methods 

All of the laboratory standard materials and chemicals were purchased from Merck (Darmstadt, Germany), Thermo Scientific (Waltham, MA, USA), VWR International (Darmstadt, Germany) and SERVA Electrophoresis GmbH (Heidelberg, Germany), if not otherwise stated.

### 4.1. Animal Model

We conducted the mouse model modified according to Yoshimoto, Loo et al., to induce hepatocellular carcinoma that is based on the development of a non-alcoholic steatohepatitis (NASH) [47]. Therefore, male C57BL/6 mice were treated with 50 µL 0.5 % DMBA (7,12-Dimethylbenz[a]anthracene) dissolved in acetone on the fourth or fifth postnatal day. With four weeks of age, these mice were fed western diet (40 kcal% fat, 20 kcal% fructose, and 2 kcal% cholesterol, Research Diets, Inc., New Brunswick, NJ, USA) for 30 weeks. With 34 weeks of age, in vivo μCT imaging was performed while using a gantry-based flat-panel microcomputed tomography scanner (TomoScope Duo, CT Imaging, Erlangen, Germany). The mice were scanned before and immediately after *i.v.* injection of Imeron^®^ 400 MCT (BRACCO Imaging, Konstanz, Germany, max. 60 mg/25 g body weight into the tail vein), a non-ionic iodized radio-opaque substance. The animals were anesthetized by inhalation anesthesia in an anesthetic chamber (Drägerwerk AG, Lübeck, Germany, 2 Vol% isoflurane in O_2_ enriched air) during the entire in vivo imaging process. For each mouse, a CT scan was performed at 54 kV (0.9 mA), acquiring 1440 projections of size 1944 × 1536 over 9 min. of continuous rotation or at 75 kV (120 µA), 2 × 2 detector binning, 1024 projections over 360 degrees, 1.3 times magnification of continuous rotation. A Feldkamp-type reconstruction algorithm (CT-Imaging, Erlangen, Germany) was implemented, including ring artifact correction. The reconstructed data were visualized and analyzed with Imalytics Preclinical software. Livers were segmented while using an automated segmentation method with interactive correction of segmentation errors. NASH was confirmed by the use of NAS and SAF Score. The animal experiments were approved by the governmental care and use committee (LANUV, Recklinghausen, NRW, Germany) and they were conducted in accordance with the federal German law and European directive 2010/63/EU on the protection of animals used for scientific procedures.

### 4.2. Isolation of Primary Mouse Hepatocytes

Isolation of N-HCC25 was conducted as described by Hesse, Jascke et al. [91].

### 4.3. Cell Culture and Stimulation 

N-HCC25 cells were cultured in Dulbecco’s modified Eagle’s (DMEM) containing 10 % FBS, 4.5 g/L D-glucose, 4 mM glutamine, and 1 % penicillin/streptomycin (full medium [FM]) at 37 °C in a humidified 5 % CO_2_ atmosphere. All of the cell culture plastics used in subsequent experiments were coated with collagen A (0.1 mg/mL). The wash steps were conducted with PBS. Cells were detached by incubation with 2 mL trypsine/EDTA (0.05 %/0.02 % in PBS, without Ca^2+^/Mg^2+^) for 20 min. at 37 °C. Cells stained with 0.4% trypan blue were counted under light microscope (DM IL LED Fluo, Leica Microsystems CMS GmbH, Wetzlar, Germany) in a Neubauer chamber. Pictures of cell culture were taken with DFC345 FX from Leica. First generation (Everolimus) and second generation mTOR inhibitors (KU-0063794) dissolved in DMSO (Applichem, Darmstadt, Germany) were applied (Absource Diagnostics GmbH, Munich, Germany) to test the influence of mTOR on cell proliferation of N-HCC25 cells. Concentrations and incubation times are provided in the materials and methods section of each experiment. The control cells were cultured in full medium (FM control) or received DMSO as vehicle treatment (DMSO control). 

### 4.4. Cytogenetic Analysis

Structural and numerical alterations of N-HCC25 cells (2.5 × 10^4^) were explored by conventional karyotyping via GTG-banding. Metaphase spreads were obtained according to the standard protocols of hypotonic treatment (0.56% KCl) and Carnoy’s fixation (methanol/acetic acid, 3:1). GTG-banding was made by means of trypsin pretreatment for the partial removal of DNA-associated (chromosomal) proteins and subsequent Giemsa staining while using standard procedures. Microscopy was performed with Axioplan fluorescence microscope (Carl Zeiss, Jena, Germany) and IKARUS^TM^ and ISIS^TM^ digital imaging systems (MetaSystems, Altlussheim, Germany). 20 and 26 GTG banded metaphases were analysed per passage, respectively. Whole chromosome painting (wcp) probes for fluorescence in situ hybridisation (FISH)-based visualisation of x- and y-chromosomes were purchased from Cytocell Ltd. (Cambridge, UK) and applied as recommended by the manufacturer.

### 4.5. PCR

Total RNA was isolated from 1 × 10^6^ N-HCC25 cells of diverse passages using peqGOLD RNAPure™ (VWR Peqlab, Darmstadt, Germany), according to the manufacturer’s recommendations. The concentration and purity of total RNA was measured with a Synergy HT reader (Biotek Instruments, Winooski, VT, USA). cDNA synthesis was carried out with 2 µg total RNA. gDNA was digested by DNaseI treatment and cDNA was synthesized with Maxima RT while using oligo(dt)18 and random hexamer mixed priming. PCR was performed with 20 ng cDNA in a Veriti™ 96-well Thermal Cycler with a standard PCR cycle protocol (Applied Biosystems, Darmstadt, Germany). Table 1 lists the information regarding primer sequence, annealing temperature, and amount of cycle per run. GAPDH served as reference gene. Gel electrophoresis was conducted while using 2% agarose gel containing 2 µL Midori Green Advance (Biozym Scientific GmbH, Wien, Austria) at 140 Volt for 40 min. (PowerPac Basic, Bio-rad Laboratories, Hercules, CA, USA). 100bp DNA ladder served as reference. Liver samples of DDC treated mice (DDC), a valuable model for stem cell activation in the liver, and untreated primary hepatocytes (PH) were used as positive controls for marker expressions. No template control (NTC) served as the negative control.

### 4.6. Western Blot

5 × 10^5^ N-HCC25 cells were seeded on cell culture dish (100 × 20mm) for 42 or 24 h, respectively, in full medium prior to the treatments. Subsequently, the cells were stimulated for 6 or 24 h respectively with 2.5 µM Everolimus or KU-0063794. In order to ensure mTOR activity, a confluence of 80 % was not exceeded. After 48 h in total, proteins were isolated by sonification (6 cycles of 20 s with 10 % power, Sonopuls, Bandelin, Berlin, Germany) in RIPA buffer. Lowry Assay (DC Protein Assay, Bio-Rad Laboratories, Hercules, CA, USA) determined the protein concentration, according to the manufacturer’s recommendations while using the Synergy HT reader. Electrophoresis was conducted using precast gels (ServaGel TG Prime 4–12%) at 50 V for 20 min. and subsequently 120 Volt for 160 min. Triple Color Standard III served as protein ladder. Immunoblot was carried out at 70 V for 135 min. Blocking was conducted with milk or BSA for one hour at room temperature under continuous rocking. The primary antibodies were incubated at 4 °C over night and secondary antibodies for one hour at room temperature. Table 3 and Table 4 shows all applied antibodies, including target size, order number, dilution, and required blocking solution. The membranes were incubated with Western Lightning Plus ECL (PerkinElmer Inc., MA, USA) for visualization. The scan was conducted with ChemoCam (ECL-Imager, Intas Science Imaging Instruments GmbH, Göttingen, Germany, program: Chemo Star). All of the wash steps were conducted with TBS-T. FM control and DMSO control (0.025% DMSO for 24 h) served as reference.

### 4.7. Starvation Experiments

1 × 10^5^ N-HCC25 cells were seeded in six well culture plates in a total volume of 2 mL full medium. After 24 h, basic nutrients were reduced for 24 or 48 h. respectively, including the following concentrations: glutamine (0, 0.1, 1, or 2 mM), FBS (0%, 1%, 2.5%, or 5%), and glucose (0, 0.5, 1, or 2 g/L). Cells that were cultured in full medium served as control. 

### 4.8. RTCA (Real-Time Cell Analysis)

1 × 10^3^ or 4 × 10^3^ N-HCC25 cells, respectively, were seeded in a collagen-coated 96-well E-plate of the xCELLigence system in 200 µL full medium, according to the manufacturer’s instructions under normal culture conditions (xCELLigence^®^ RTCA SP instrument, Roche Diagnostics, Mannheim, Germany). After an adherence phase of 24 h, 100 µL of the medium were exchanged. 48 h after seeding, the cells were either starved (5% FBS, 2mM glutamine, or 2.25 g/L glucose) or treated with Everolimus or KU-0063794 (1, 2.5, or 5 µM). The impedance was measured by RTCA software version 1.2.1 at an interval of 15 min. for 105 h and subsequently at an interval of one hour until the end of the experiment. The results are given as cell index (CI) for time course depiction and endpoint analyses of pre-defined timepoints or as slope (1/h) for pre-defined growth phases. The data of nutrient reduction were normalized to the timepoint of stimulation. FM and DMSO control (0.025% DMSO for 24 h) served as reference.

### 4.9. Flow Cytometry

Two-dimensional cell cycle analysis was conducted to explore the effect of mTOR inhibition on cell cycle distribution by treatment with BrdU (5-bromo-2’-deoxyuridine, Applichem, Darmstadt, Germany) and PI (propidium iodide). While PI is incorporated in the entire DNA, the thymidine analogue BrdU is only contained in de novo synthesized DNA. Thus, the cells can be referred to different stages of cell cycle (G_1_/G_0_ phase: PI^low^BrdU^low^, S phase: PI^+^BrdU^high^; G_2_/M phase: PI^high^BrdU^low^). Initially, 1 × 10^6^ N-HCC25 cells were cultured in full medium for 24 h and subsequently incubated with Everolimus or KU-0063794 (2.5 µM or 5 µM) for 24 h. After pulse-labeling with 10 µM BrdU for 30 min. at 37°C, the cells were immediately harvested for BrdU/PI staining. First, the cells were fixated with 70% ethanol at −20°C and incubated with 2 N HCl (Applichem, Darmstadt, Germany) containing 0.5% Triton X-100 at room temperature for 30 min. each. After neutralising with 0.1 M Na_2_B_4_O_7_ at pH 8.5, cell count was adjusted to 1 million/mL in PBS with 1% BSA and 0.05% Tween-20. After incubation with anti-BrdU antibody (mouse, clone Bu20a, Agilent, Santa Clara, CA, USA; 1:100 in PBS with 1% BSA) at 4 °C overnight, an Alexa Fluor 488-conjugated secondary antibody (Goat anti-mouse IgG; 1:200 in PBS with 1% BSA) was added for 30 min. at room temperature, being protected from light. Finally, the cells were stained with PI (10 µg/mL in PBS). Each step was followed by a wash step with PBS with 1% BSA. FM and DMSO control (0.05% DMSO) served as reference. Samples were quantified with a BD LSRFortessa Cell Analyzer (BD Biosiences, San Jose, CA, USA). FlowJo Software Version 10 (FlowJo LLC, Ashland, Oregon, USA) was used for data analysis. The applied gating strategy and technical controls are provided in the Appendix A.

### 4.10. Statistical Analysis

The Bartlett test was used to check for variance homogeneity. Normal distribution was tested with the Shapiro–Wilk test. Multiple comparisons were analyzed by one- or two-way ANOVA followed by Bonferroni’s or Tukey’s multiple comparison post-hoc tests, as indicated in the figure legends. Data represent mean ± SEM if not otherwise stated. All statistical analyses and graphs were created with GraphPad Prism 7.0 (GraphPad; La Jolla, USA) and JMP 10.0 Software (SAS Institute Inc., NC, USA). *p* values < 0.05 were considered as statistically significant.

## Figures and Tables

**Figure 1 ijms-20-05658-f001:**
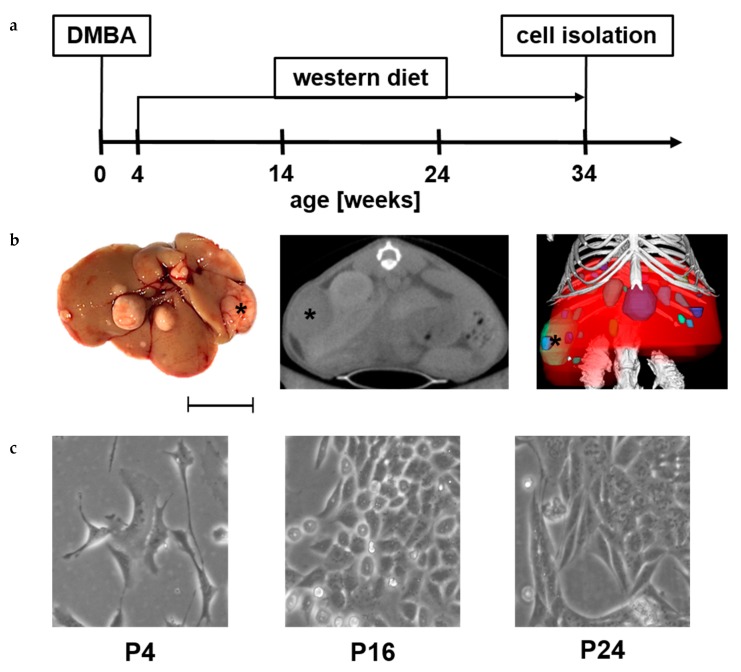
Isolation of N-HCC25 cell line. (**a**) Male C57BL/6 mice were treated with 7,12-Dimethylbenz[a]anthracene (DMBA) on the 4th postnatal day and fed western diet for 30 weeks to develop hepatocellular cancer in NASH; (**b**) Characteristic picture of liver tissue that was used for isolation of N-HCC25 (view from dorsal, scale = 1 cm). Two-dimensional (2D) cross-sectional µCT image (transversal) of the mouse. 3D volume renderings of segmented bones (white), liver (red) and tumors (different colour for each tumor) upon in vivo µCT imaging (tumor marked with black asterisk, “R” marks right side of liver or mouse, scale = 1 cm); (**c**) Representative pictures from early and late passages of N-HCC25 (scale = 50 µm).

**Figure 2 ijms-20-05658-f002:**
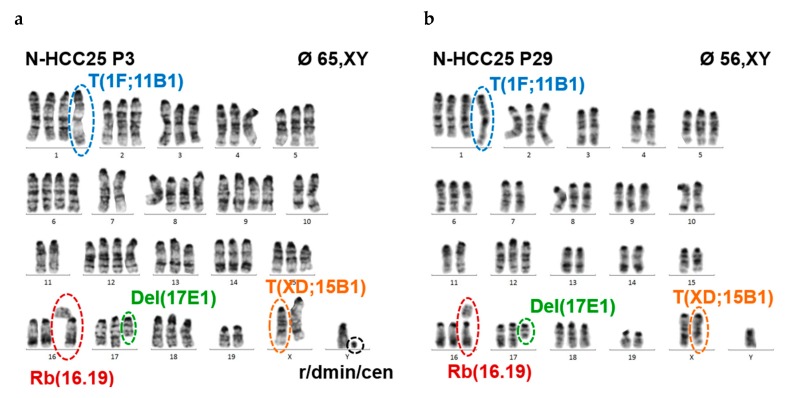
N-HCC25 cells display structural and numerical chromosomal aberrations associated with HCC pathology. Karyograms of N-HCC25 cells, (**a**) early passage 3 and (**b**) late passage 29. Fluorescence in situ hybridisation (FISH, whole-chromosome-painting probe X for X-chromosomal DNA content of early (**c**) and late (**d**) passage (magnification = 1200×). T = unbalanced translocation; Rb = Robertsonian Fusion; Del = Deletion; Hc (r/dmin/cen) = pericentric heterochromatin; wcp = whole chromosome painting probe. Detailed information about aberration are shown in brackets.

**Figure 3 ijms-20-05658-f003:**
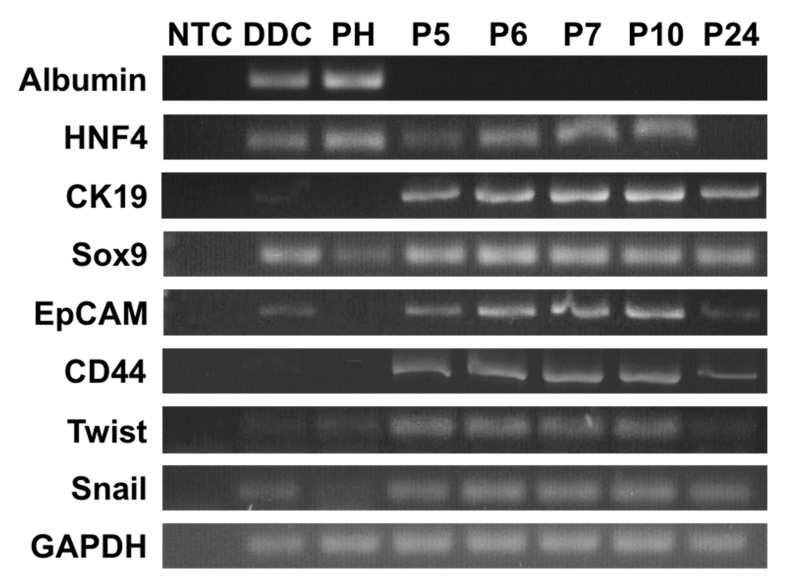
N-HCC25 cells derived from hepatocytes and express tumor-like gene markers. RNA was isolated and transcribed in cDNA for various passages (P) of N-HCC25 cells. PCR was conducted with specific markers of hepatocytes (Albumin, HNF4), cancer stem cells (CK19, Sox9, EpCAM, CD44) and epithelial-mesenchymal transition (Twist, Snail). GAPDH served as reference gene. No template control (NTC) served as negative control; liver samples from DDC treated mice (DDC) and primary hepatocytes (PH) served as positive controls.

**Figure 4 ijms-20-05658-f004:**
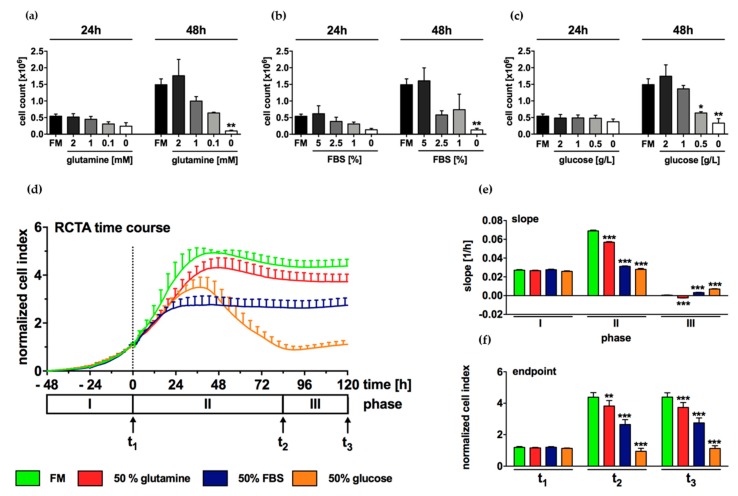
Growth of N-HCC25 cells depends mostly on sufficient levels of glucose and FBS. (**a**–**c**) After an adherence phase of 24 h, N-HCC25 cells were cultured with reduced glutamine, FBS or glucose in various concentrations for 24 and 48 h. Subsequently, cells were counted in Neubauer chamber. (**d**–**f**) Starvation experiments were repeated in xCELLigence real-time cell analyzer (RTCA) for longitudinal measurement of cell density. 48 h after seeding in full medium (FM), cells were treated with halved amounts of glutamine (2 mM), FBS [5 %] or glucose [2.25 g/L] for 120 h. Data were normalized to the timepoint of stimulation as indicated by the dashed line and grouped into three characteristic phases (I–III). Each phase ends at a respective timepoint t_1–3_. (**e**) Slope and (**f**) endpoint were calculated from the data shown in (**d**). Cells treated with full medium (FM) containing 4 mM glutamine, 10 % FBS and 4.5 g/L glucose served as control (**a**–**f**). Data represent mean ± SEM; n ≥ 3 (starvations of glutamine, FBS & glucose n = 3; FM24h *n* = 9 and FM48h *n* = 10) (**a**–**c**); or mean ± SD; *n* (FM) = 4, *n* (glutamine) = 8, *n* (FBS) = 6, *n* (glucose)= 7 (**d**–**f**). Statistical significances are indicated as * *p* < 0.05, ** *p* < 0.01, *** *p* < 0.001 vs. FM control in same phase or timepoint. One-way ANOVA (**a**–**c**) or two-way ANOVA respectively (**d**–**f**) with Bonferroni multiple comparison post-hoc test, GraphPad Prism 7 software, La Jolla, CA, USA).

**Figure 5 ijms-20-05658-f005:**
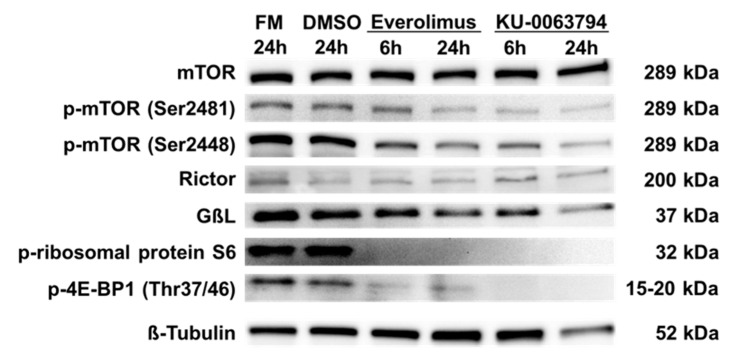
N-HCC25 cells exhibit mTOR activity, which can be blocked by specific inhibitors. The role of mTOR protein pathway in N-HCC25 was analyzed in Western blot including central proteins of mTOR complex 1 and 2 signaling. 42 h (24 h) after seeding in full medium (FM), N-HCC25 cells were treated with 2.5 µM Everolimus or KU-0063794 for 6 h (24 h). Cells cultured in FM with or without 0.025% DMSO served as controls.

**Figure 6 ijms-20-05658-f006:**
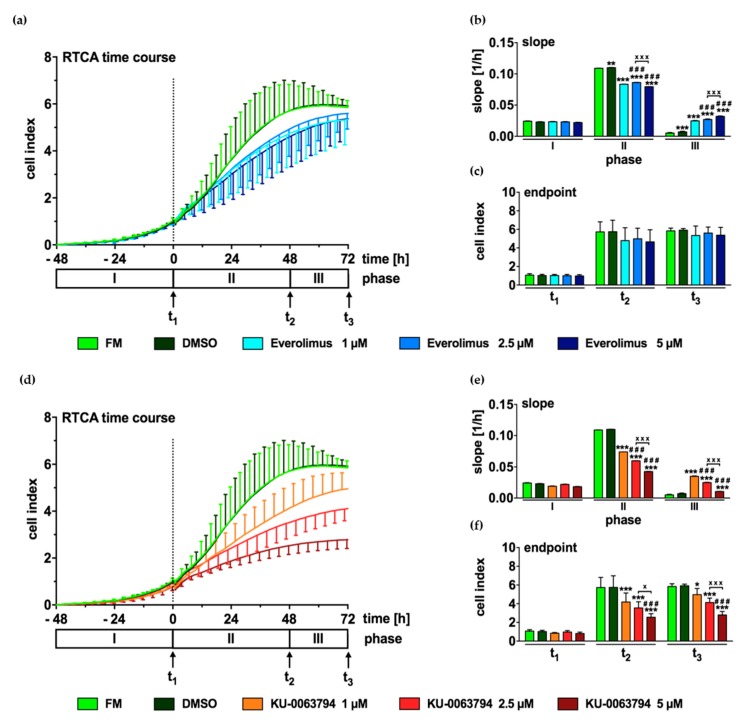
mTOR inhibition leads to decreased proliferation of N-HCC25 cells. Real-time cell analyzer (RTCA) was used to compare the effect of first generation (Everolimus) and second generation (KU-0063794) mTOR inhibition on cell proliferation. 48 h after seeding, N-HCC25 cells were treated with 1 µM, 2.5 µM, or 5 µM Everolimus (**a**–**c**) or same concentrations of KU-0063794 (**d**–**f**). Cell proliferation was recorded for 120 h in total and grouped into three characteristic phases (I-III). Each phase ends at a timepoint t_1–3_. Slope (**b**,**e**) and endpoint (**c**,**f**) were calculated from the data shown in (**a**) and (**d**). Graphs represent mean ± SD; *n* = 8 (KU-0063794 2.5 µM, *n* = 7). Statistical significances are indicated as * *p* < 0.05, ** *p* < 0.01, *** *p* < 0.001 vs. full medium (FM) and DMSO in same phase or timepoint ### *p* < 0.001 vs. 1 µM Everolimus or KU-0063794 respectively; x *p* < 0.05, xxx *p* < 0.001 as indicated. Two-way ANOVA with Tukey’s multiple comparison post-hoc test, GraphPad Prism 7 software, La Jolla, CA, USA.

**Figure 7 ijms-20-05658-f007:**
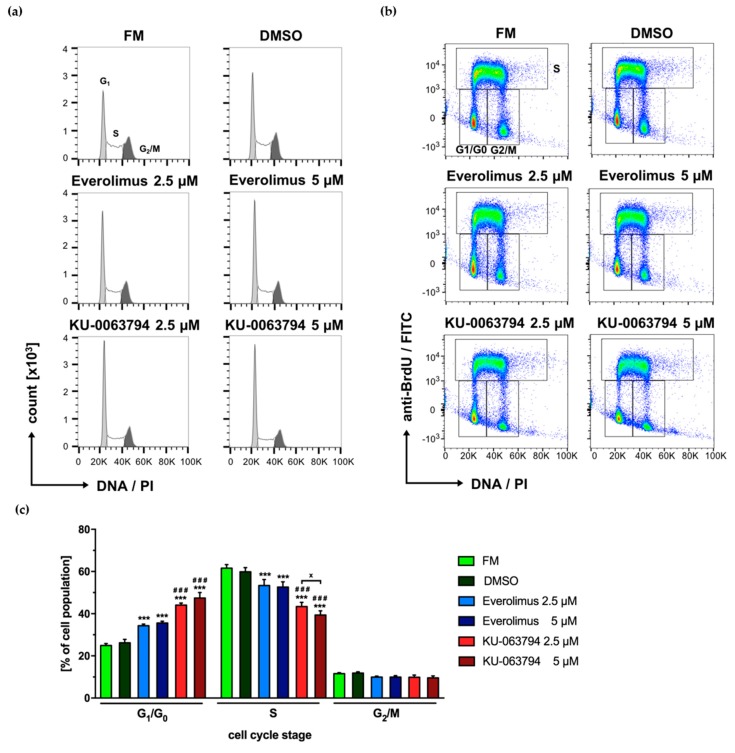
mTOR pathway inhibition drives N-HCC25 cells into G_1_/G_0_ cell cycle arrest. After an adherence phase of 24 h, N-HCC25 cells were treated with Everolimus or KU-0063794 (2.5 µM or 5 µM each) for 24 h and analyzed by flow cytometry for incorporation of BrdU and PI. The applied gating strategy is shown in Appendix A. Cells cultured in full medium (FM) with or without 0.05% DMSO served as controls. Representative data are shown in the form of histograms (**a**) and dot plots (**b**). Statistical analysis of cell cycle distribution obtained from (**b**) is shown in (**c**). Graphs represent mean ± SD; *n* (FM, DMSO) = 3, *n* (inhibitors) = 5. Statistical significances are indicated as *** *p* < 0.001 vs. FM and DMSO control in the same cell cycle stage; ### *p* < 0.001 vs. Everolimus 2.5 µM and 5 µM in same cell cycle stage; x *p* < 0.05, as indicated. (Two-way ANOVA with Tukey multiple comparison post hoc test, GraphPad Prism 7.0 software, La Jolla, CA, USA).

**Table 1 ijms-20-05658-t001:** Numerical aberrations identified by cytogenetic approaches. Frequency of aneuploidies (tetrasomy and reduction to disomy) are shown in the upper part and general ploidy grades of investigated passages (P3 & P29) in the lower part of the table.

Chromosome	Aneuploidy (4×)		Loss of Additional Chromosomes (2x)	
	P3	P29	P3	P29
2	95%	4%	-	8%
3	60%	-	10%	50%
4	30%	4%	10%	85%
5	75%	-	-	15%
6	45%	54%	-	4%
7	50%	4%	20%	88%
8	70%	38%	-	-
9	55%	4%	5%	31%
10	35%	-	15%	81%
11	25%	-	-	-
12	65%	19%	-	8%
13	20%	-	5%	38%
14	30%	-	-	58%
15	15%	8%	-	8%
16	25%	19%	-	11%
17	5%	-	-	8%
18	-	-	50%	46%
19	15%	15%	15%	4%
	ploidy grade	#chromosomes	range	
P3	hypo-tetraploid	65	60–69	
P29	hypo-triploid	56	50–59	

**Table 2 ijms-20-05658-t002:** Structural aberrations identified by cytogenetic approaches. Frequency of structural aberrations and chromosomal rearrangements. T = unbalanced translocation; Rb = Robertsonian Fusion; Del = Deletion; r/dmin/cen = pericentric heterochromatin.

Kind of Aberration	P3	P29
T (1F;11B1)	96%	100%
Rb 16.19	73%	86%
Del(17E1)	92%	100%
T (XD;15B1)	82%	93%
loss of Y	44%	55%
r/dmin	48%	61%

**Table 3 ijms-20-05658-t003:** List of primers for polymerase chain reaction (PCR).

Primer		Sequence	Annealing Temperature	Cycles
Albumin	for rev	TCCTGGGCACGTTCTTGTAT TGCTTTCTGGGTGTAGCGAA	58.5 °C	30
HNF4	for rev	AAGGTGCCAACCTCAATTCATC CACATTGTCGGCTAAACCTGC	60 °C	30
CK19	for rev	GTCCTACAGATTGACAATG CACGCTCTGGATCTGTGACAG	57 °C	30
Sox9	for rev	GTGAAGAACGGACAAGCGGA GATTGCCCAGAGTGCTCGC	60 °C	40
EpCAM	for rev	CGGCTCAGAGAGACTGTGTC GATCCAGTAGGTCCTCACGC	57.5 °C	30
CD44	for rev	CAGAGGCGACTAGATCCCTC GAGTCACAGTGCGGGAACTC	59 °C	30
Twist	for rev	GCCGGAGACCTAGATGTCATTG CCACGCCCTGATTCTTGTGA	60 °C	40
Snail	for rev	TCTGCACGACCTGTGGAAAG GTTGGAGCGGTCAGCAAAAG	60 °C	40
GAPDH	for rev	AGGTCGGTGTGAACGGATTTG TGTAGACCATGTAGTTGAGGTCA	60 °C	40

**Table 4 ijms-20-05658-t004:** List of antibodies for Western blot.

Antibody	Target Size	Order Number	Dilution	Blocking Solution
mTOR	289 kDa	CST 2983	1:500	milk powder
p-mTOR (Ser2481)	289 kDa	CST 2974	1:1000	BSA
p-mTOR (Ser2448)	289 kDa	CST 5536	1:500	BSA
Rictor	200 kDa	CST 2114	1:2500	BSA
GßL	37 kDa	CST 3274	1:1000	BSA
p-ribosomal protein S6	32 kDa	SC 293144	1:200	BSA
p-4E-BP1	15–20 kDa	CST 2855	1:1000	BSA
ß-Tubulin	52 kDa	PA5 16863	1:1000	milk powder
Goat-anti-mouse	-	CST 7076	1:1000	-
Goat-anti-rabbit	-	CST 7074	1:2500	-

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
