# Peer review of "A Newly Established Murine Cell Line as a Model for Hepatocellular Cancer in Non-Alcoholic Steatohepatitis"

_ijms, 2019, doi:10.3390/ijms20225658_

Round 1

Reviewer 1 Report

This is an interesting article on a new murine-derived cell line, for the study of hepatocellular carcinoma in non-alcoholic steatohepatitis (NASH). Chromosomal aberrations, specific markers (HNF4, Twist, Snail, CD44, EpCAM,
CK19, Sox9), dependency on nutrients (glucose, fetal bovine serum and glutamine) have been thoroughly evaluated and clearly described. Despite the above, another focus was on the inhibition of mTOR, by first and second generation drugs. The N-HCC25 cells expressed simultaneously hepatocitic and cancer stem cell markers. Proliferation dipended highly on glucose and fetal bovine serum but not glutamine. There was observed an important decrease of cell growth due to the use of second generation mTOR inhibitor.

I find the article interesting, useful and well written.

On line 297 to should be too

Author Response

We thank the reviewer for the kind words. We corrected the mentioned spelling on line 297 as suggested.

Reviewer 2 Report

I carefully reviewed the manuscript ID: ijms-635656, entitled " A newly established murine cell line as a model for 2 hepatocellular cancer in non-alcoholic steatohepatitis " by Andreas Kroh et alAuthors established and characterized a new murine model cell line, N-HCC25, for hepatocellular cancer (HCC) in non-alcoholic steatohepatitis (NASH). In general, the study was well designed, methodology was appropriate, and the results and interpretation were scientifical soundness. Although the presentation is impressive, a few points need more clarification.

General concerns:

If authors claimed that the N-HCC25 may serve as a suitable in vitro model for NASH-derived HCC, they should offer the following data: The difference and resemblance between N-HCC25 and human NASH-derived HCC. If no human NASH-derived HCC cell line is available, live liver tumor biopsy from NASH patients is an alternative. The difference and resemblance between N-HCC25 and human HBV or HCV-derived HCC In background, authors should provide the criteria used to define NASH-derived HCC. In addition to mTOR pathway, author should provide another possible pathway(s) that might cause HCC development in NASH and viral-induced HCC.

Is the N-HCC25 cell line able to grow in a treatment-naïve mice?

Author Response

I carefully reviewed the manuscript ID: ijms-635656, entitled " A newly established murine cell line as a model for 2 hepatocellular cancer in non-alcoholic steatohepatitis " by Andreas Kroh et al. Authors established and characterized a new murine model cell line, N-HCC25, for hepatocellular cancer (HCC) in non-alcoholic steatohepatitis (NASH). In general, the study was well designed, methodology was appropriate, and the results and interpretation were scientifical soundness. Although the presentation is impressive, a few points need more clarification.

General concerns:

If authors claimed that the N-HCC25 may serve as a suitable in vitro model for NASH-derived HCC, they should offer the following data:

Response: General remark:

We would like to thank the reviewer for the many stimulating and constructive remarks. We appreciate the encouraging comments and critical evaluation of the relevance of our data and stated concerns, which we carefully considered point-by-point. Our detailed responses are shown below.

Since the given timeframe for this revision (5 days) did not allow us to generate additional experimental data, we addressed the arisen issues by additional representative data from our mouse model as well as further references and discussion.

Point 1: The difference and resemblance between N-HCC25 and human NASH-derived HCC. If no human NASH-derived HCC cell line is available, live liver tumor biopsy from NASH patients is an alternative.

Response: We thank the reviewer for this advice. To address this point we included a passage in the introduction regarding pathological changes in human NASH. Furthermore, we added representative images of H&E, PAS as well as Sirius Red staining of liver sections out of our NASH-HCC model in the supplements. Based on these stainings we assessed the NAS and SAF scores, which include NASH-specific hallmarks such as ballooning of hepatocytes, steatosis, inflammation and fibrosis and serve for diagnosis of NASH in humans. Since the remaining data of the animal experiment (which also comprises KO studies) is planned to be published separately, we are not able to provide more data on this point.

Point 2: The difference and resemblance between N-HCC25 and human HBV or HCV-derived HCC

Response: We included the following passage in the introduction that describes the pathogenesis of HBV/HCV as well as NASH-derived HCC, thereby highlighting the differences between them:

“By escaping the HLA-II-mediated immune response, HBV/HCV infects hepatocytes, whereby oxidative stress and subsequent recruitment of inflammatory cells activate hepatic stellate cells (HSC) and collagen deposition. Besides, several HCV proteins are also known to induce fibrogenic and inflammatory processes of HSCs [13,14]. In viral hepatitis, initial fibrosis mostly develops within portal tracts, leading to persistent portal inflammation, cirrhosis and ultimately HCC. In contrast, dyslipidaemia, a commonly observed phenomenon in obese patients suffering from non-alcoholic fatty liver disease (NAFLD) leads to increased hepatic lipid deposition (steatosis) and more generalized liver inflammation. This in turn, triggers a copious development of liver fibrosis, causing NASH that can advance to HCC either via cirrhosis or without. [15-18]”

Since our model exhibits similar pathological changes as in human NASH-HCC (see response to the previous point), these differences are also true between our model and HBV/HCV-derived HCC.

Point 3: In background, authors should provide the criteria used to define NASH-derived HCC.

Response: In our mouse model, mice were treated with DMBA and a Western Diet to induce NASH-derived HCCs. DMBA is a carcinogen and the Western Diet induces NASH. As mentioned above, we used the NAS and SAF scores for defining NASH and thereby conclude that observed HCCs developed from NASH. While DMBA-treated mice fed a Western diet developed NASH (NAS4, S2A2F2) and HCCs, control mice treated with DMBA and fed a normal diet did not show NASH (NAS1, S0A1F0) and did not develop liver tumors.

Point 4: In addition to mTOR pathway, author should provide another possible pathway(s) that might cause HCC development in NASH and viral-induced HCC.

Response: As HCC is a heterogeneous tumor, various dysfunctions of cellular processes and signalling pathways have been identified to be involved in its development. These include alterations of telomerase reverse transcriptase (TERT), the Wnt pathway, JAK-STAT pathway, DNA repair mechanisms by TP53, chromatin remodelling by SWI/SNF as well as the PI3K-AKT-mTOR pathway, all of which are investigated as possible targets for anticancer therapy. Since metabolic reprogramming is a common and significant pathological change in HCC and cancer in general, the mTOR pathway is of specific interest. Especially since mTOR inhibitors already found their way into clinical application. However, first-generation inhibitors mostly failed to improve survival of HCC patients. For this reason, we chose this pathway to investigate the response of N-HCC25 cells to mTOR inhibition. Interestingly, we found out, that treatment with a second-generation mTOR inhibitor (KU-0063794) resulted in a strong decrease of cell growth in a dose dependent manner while a first-generation mTOR inhibitor (Everolimus) in contrast reduced cell proliferation only slightly.

We included a passage in the discussion section to address this issue.

Point 5: Nevertheless, we added a brief passage in which we address this point.

Is the N-HCC25 cell line able to grow in a treatment-naïve mice?

Response: Unfortunately, we did not use these cells in orthotopic models yet, but we thank the reviewer for this helpful suggestion. We will keep that in mind for the upcoming studies with these cells.